# Plan Arithmetic: Compositional Plan Vectors for Multi-Task Control

**Coline Devin**   **Daniel Geng**   **Pieter Abbeel**   **Trevor Darrell**   **Sergey Levine**

University of California, Berkeley

## Abstract

Autonomous agents situated in real-world environments must be able to master large repertoires of skills. While a single short skill can be learned quickly, it would be impractical to learn every task independently. Instead, the agent should share knowledge across behaviors such that each task can be learned efficiently, and such that the resulting model can generalize to new tasks, especially ones that are compositions or subsets of tasks seen previously. A policy conditioned on a goal or demonstration has the potential to share knowledge between tasks if it sees enough diversity of inputs. However, these methods may not generalize to a more complex task at test time. We introduce compositional plan vectors (CPVs) to enable a policy to perform compositions of tasks without additional supervision. CPVs represent trajectories as the sum of the subtasks within them. We show that CPVs can be learned within a one-shot imitation learning framework without any additional supervision or information about task hierarchy, and enable a demonstration-conditioned policy to generalize to tasks that sequence twice as many skills as the tasks seen during training. Analogously to embeddings such as word2vec in NLP, CPVs can also support simple arithmetic operations – for example, we can add the CPVs for two different tasks to command an agent to compose both tasks, without any additional training.

## 1   Introduction

A major challenge in current machine learning is to not only interpolate within the distribution of inputs seen during training, but also to generalize to a wider distribution. While we cannot expect arbitrary generalization, models should be able to compose concepts seen during training into new combinations. With deep learning, agents learn high level representations of the data they perceive. If the data is drawn from a compositional environment, then agents that model the data accurately and efficiently would represent that compositionality without needing specific priors or regularization. In fact, prior work has shown that compositional representations can emerge automatically from simple objectives, most notably a highly structured distribution such as language. These techniques do not explicitly *train* for compositionality, but employ simple structural constraints that lead to compositional representations. For example, Mikolov et al. found that a language model trained to predict nearby words represented words in a vector space that supported arithmetic analogies: "king" -"man" + "woman" = "queen" [29]. In this work, we aim to learn a compositional feature space to represent robotic skills, such that the addition of multiple skills results in a plan to accomplish all of these skills.

Many tasks can be expressed as compositions of skills, where the same set of skills is shared across many tasks. For example, assembling a chair may require the subtask of picking up a hammer, which is also found in the table assembly task. We posit that a task representation that leverages this compositional structure can generalize more easily to more complex tasks. We propose learning an

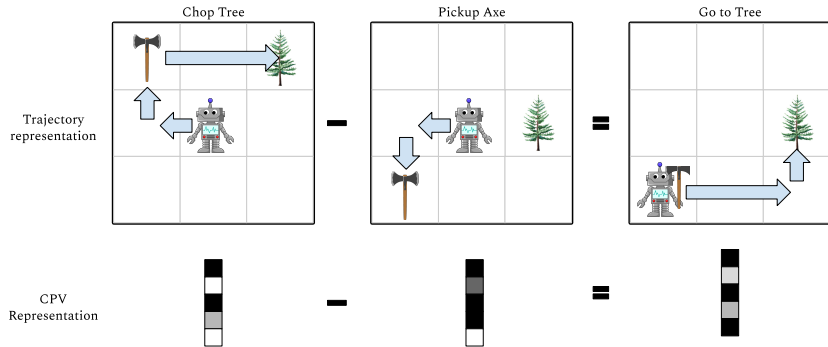

Figure 1: Compositional plan vectors embed tasks into a space where adding two vectors represents the composition of the tasks, and subtracting a sub-task leaves an embedding of the remaining sub-tasks needed for the task.

embedding space such that tasks could be composed simply by adding their respective embeddings. This idea is illustrated in Figure 2.

In order to learn these representations without additional supervision, we cannot depend on known segmentation of the trajectories into subtasks, or labels about which subtasks are shared between different tasks. Instead, we incorporate compositionality directly into the architecture of the policy. Rather than conditioning the policy on the static embedding of the reference demonstration, we condition the policy on the *difference* between the embedding of the whole reference trajectory and the partially completed trajectory that the policy is outputting an action for.

The main contributions of our work are the compositional plan vector (CPV) representation and a policy architecture that enables learning of CPVs without any sub-task level supervision. CPVs enable policies to generalize to significantly longer tasks, and they can be added together to represent a composition of tasks. We evaluate CPVs in the one-shot imitation learning paradigm [11, 12, 19] on a discrete-action environment inspired by Minecraft, where tools must be picked up to remove or build objects, as well as on a 3D simulated pick-and-place environment.

## 2 Related Work

For many types of high dimensional inputs, Euclidean distances are often meaningless in the raw input space. Words represented as one-hot vectors are equally distant from all other words, and images of the same scene may have entirely different pixel values if the viewpoint is shifted slightly. This has motivated learning representations of language and images that respect desirable properties. Chopra et al. [3] showed that a simple contrastive loss can be used to learn face embeddings. A similar method was also used on image patches to learn general image features [35]. Word2vec found that word representations trained to be predictive of their neighbor words support some level of addition and subtraction [29, 24, 21]. More recently, Nagarajan used a contrastive approach in learning decomposable embeddings of images by representing objects as vectors and attributes as transformations of those vectors [30]. These methods motivate our goal of learning an embedding space over tasks that supports transformations such as addition and subtraction. Notably, these methods don't rely on explicit regularization for arithmetic operations, but rather use a simple objective combined with the right model structure to allow a compositional representation to emerge. Our method also uses a simple end-to-end policy learning objective, combined with a structural constraint that leads to compositionality.

Hierarchical RL algorithms learn representations of sub-tasks explicitly, by using primitives or goal-conditioning [4, 32, 26, 13, 27, 7, 2, 39], or by combining multiple Q-functions [15, 36]. Our approach does not learn explicit primitives or skills, but instead aims to summarize the task via a compositional task embedding. A number of prior works have also sought to learn policies that are conditioned on a goal or task [22, 8, 23, 38, 5, 14, 18, 28, 6, 33], but without explicitly considering compositionality. Recent imitation learning methods have learned to predict latent intentions of demonstrations [16, 25]. In the one-shot imitation learning paradigm, the policy is conditioned on reference demonstrations at both test and train time. This problem has been explored with meta-

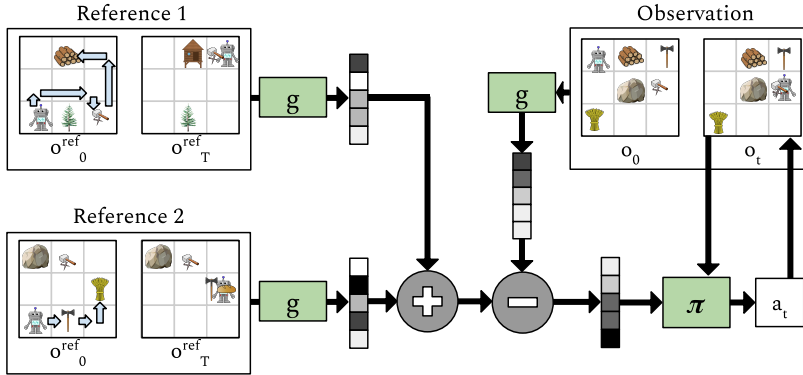

Figure 2: By adding the CPVs for two different tasks, we obtain the CPV for the composition of the tasks. To determine what steps are left in the task, the policy subtracts the embedding of its current trajectory from the reference CPV.

learning [12] and metric learning [19] for short reaching and pushing tasks. Duan et al. used attention over the reference trajectory to perform block stacking tasks [11]. Our work differs in that we aim to generalize to new compositions of tasks that are out of the distribution of tasks seen during training. Hausman et al. obtain generalization to new compositions of skills by training a generative model over skills [17]. However, unlike our method, these approach does not easily allow for sequencing skills into longer horizon tasks or composing tasks via arithmetic operations on the latent representation.

Prior methods have learned composable task representations by using ground truth knowledge about the task hierarchy. Neural task programming and the neural subtask graph solver generalize to new tasks by decomposing a demonstration into a hierarchical program for the task, but require ground-truth hierarchical decomposition during training [40, 37]. Using supervision about the relations between tasks, prior approaches have uses analogy-based objectives to learn task representations that decompose across objects and actions [31] or have set up a modular architectures over subtasks [1] or environments [9]. Unlike our approach, these methods require labels about relationships. We implicitly learn to decompose tasks without supervising the task hierarchy.

## 3    Compositional Plan Vectors

In this paper, we introduce compositional plan vectors (CPVs). The goal of CPVs is to obtain policies that generalize to new compositions of skills without requiring skills to be labeled and without knowing the list of skills that may be required. Consider a task named "red-out-yellow-in" which involves taking a red cube out of a box and placing a yellow cube into the box. A plan vector encodes the task as the sum of its parts: a plan vector for taking the red cube out of the box plus the vector for putting the yellow cube into the box should equal the plan vector for the full task. Equivalently, the plan vector for the full task minus the vector for taking the red cube out of the box should equal the vector that encodes "put yellow cube in box."

If the list of all possible skills was known ahead of time, separate policies could be learned for each skill, and then the policies could be used in sequence. However, this knowledge is often unavailable in general and limits compositionality to a fixed set of skills. Instead, our goal is to formulate an architecture and regularization that are compositional *by design* and do not need additional supervision. With our method, CPVs acquire compositional structure because of the structural constraint they place on the policy. To derive the simplest possible structural constraint, we observe that the minimum information that the policy needs about the task in order to complete it is *knowledge of the steps that have not yet been done*. That is, in the cube example above, after taking out the red cube, only the "yellow-in" portion of the task is needed by the policy. One property of this representation is that task ordering cannot be represented by the CPV because addition is commutative. If ordering is necessary to choose the right action, the policy will have to learn to decode which component of the compositional plan vector must be done first.

As an example, let $\vec{v}$ be a plan vector for the "red-out-yellow-in" task. To execute the task, a policy $\pi(\mathbf{o}_0, \vec{v})$ outputs an action for the first observation $\mathbf{o}_0$. After some number $t$ of timesteps, the policy has successfully removed the red cube from the box. This partial trajectory $\mathbf{O}_{0:t}$ can be embedded into a plan vector $\vec{u}$, which should encode the "red-out" task. We would like $(\vec{v} - \vec{u})$ to encode the remaining portion of the task, in this case placing the yellow block into the box. In other words, $\pi(\mathbf{o}_t, \vec{v} - \vec{u})$ should take the action that leads to accomplishing the plan described by $\vec{v}$ given that $\vec{u}$ has already been accomplished. In order for both $\vec{v}$ and $\vec{v} - \vec{u}$ to encode the yellow-in task, $\vec{u}$ must not encode as strongly as $\vec{v}$. If $\vec{v}$ is equal to the sum of the vectors for "red-out" and "yellow-in," then $\vec{v}$ may not encode the ordering of the tasks. However, the policy $\pi(\mathbf{o}_0, \vec{v})$ should have learned that the box must be empty in order to perform the yellow-in task, and that therefore it should perform the red-out task first.

We posit that this structure can be learned without supervision at the subtask-level. Instead, we impose a simple architectural and arithmetic constraints on the policy: the policy must choose its action based on the arithmetic difference between the plan vector embedding of the whole task and the plan vector embedding of the trajectory completed so far. Additionally, the plan vectors of two halves of the same trajectory should add up to the plan vector of the whole trajectory, which we can write down as a regularization objective for the embedding function. By training the policy and the embedding function together to optimize their objectives, we obtain an embedding of tasks that supports compositionality and generalizes to more complex tasks. In principle, CPVs can be used with any end-to-end policy learning objective, including behavioral cloning, reinforcement learning, or inverse reinforcement learning. In this work, we will validate CPVs in a one-shot imitation learning setting.

**One-shot imitation learning setup.** In one-shot imitation learning, the agent must perform a task conditioned on one reference example of the task. For example, given a demonstration of how to fold a paper crane, the agent would need to fold a paper crane. During training, the agent is provided with pairs of demonstrations, and learns a policy by predicting the actions in one trajectory by using the second as a reference. In the origami example, the agent may have trained on demonstrations of folding paper into a variety of different creatures.

We consider the one-shot imitation learning scenario where an agent is given a reference trajectory in the form of a list of $T$ observations $\mathbf{O}^{\text{ref}}_{0:T} = (\mathbf{o}^{\text{ref}}_0, ..., \mathbf{o}^{\text{ref}}_T)$. The agent starts with $\mathbf{o}_0 \sim p(\mathbf{o}_0)$, where $\mathbf{o}_0$ may be different from $\mathbf{o}^{\text{ref}}_0$. At each timestep $t$, the agent performs an action drawn from $\pi(\mathbf{a}_t | \mathbf{O}_{0:t}, \mathbf{O}^{\text{ref}}_{0:T})$.

**Plan vectors.** We define a function $g_\phi(\mathbf{O}_{k:l})$, parameterized by $\phi$, which takes in a trajectory and outputs a *plan vector*. The plan vector of a reference trajectory $g_\phi(\mathbf{O}^{\text{ref}}_{0:T})$ should encode the sequence of steps required to accomplish the goal. Similarly, the plan vector of a partially accomplished trajectory $g_\phi(\mathbf{O}_{0:t})$ should encode the steps already taken. We can therefore consider the subtraction of these vectors to encode the steps necessary to complete the task defined by the reference trajectory. Thus, the policy can be structured as

$$\pi_\theta(\mathbf{a}_t | \mathbf{o}_t, g_\phi(\mathbf{O}^{\text{ref}}_{0:T}) - g_\phi(\mathbf{O}_{0:t})), \tag{1}$$

a function parameterized by $\theta$ that takes in just the trajectory's endpoints instead of considering the full reference trajectory, and is learned end-to-end.

In this work we use a fully observable state space and only consider tasks that cause a change in the state. For example, we do not consider tasks such as lifting a block and placing it exactly where it was, because this does not result in a useful change to the state. Thus, instead of embedding a whole trajectory $\mathbf{O}_{0:t}$, we limit $g$ to only look at the first and last state of the trajectory we wish to embed. Then, $\pi$ becomes

$$\pi_\theta(\mathbf{a}_t | \mathbf{o}_t, g(\mathbf{o}^{\text{ref}}_0, \mathbf{o}^{\text{ref}}_T) - g(\mathbf{o}_o, \mathbf{o}_t)). \tag{2}$$

**Training.** With $\pi$ defined as above, we learn the parameters of the policy with imitation learning. dataset $\mathcal{D}$ containing $N$ demonstrations paired with reference trajectories is collected. Each trajectory may be a different arbitrary length, and the tasks performed by each pair of trajectories are unlabeled. The demonstrations include actions, but the reference trajectories do not. In our settings, the reference trajectories only need to include their first and last states. Formally,

$$\mathcal{D} = \{(\mathbf{O}^{\text{ref}^i}_{[0,T^i]}, \mathbf{O}^i_{[0:H^i]}, \mathbf{A}^i_{[0:H^i-1]})\}_{i=1}^N,$$

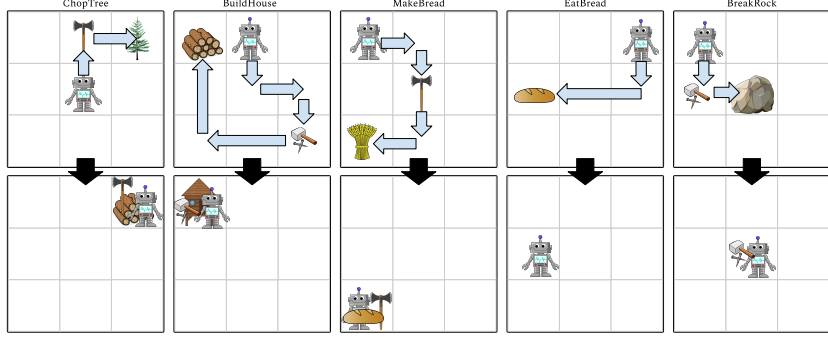

Figure 3: Illustrations of the 5 skills in the GridCraft environment. To ChopTree, the agent must pick up the axe and bring it to the tree, which transforms the tree into logs. To BuildHouse, the agent picks up the hammer and brings it to logs to transform them into a house. To MakeBread, the agent brings the axe to the wheat which transforms it into bread. The agent eats bread if it lands on a state that contains bread. To BreakRock, the agent picks up a hammer and destroys the rock.

where $T^i$ is the length of the $i^{\text{th}}$ reference trajectory and $H^i$ is the length of the $i$th demonstration. Given the policy architecture defined in Equation 1, the behavioral cloning loss for a discrete action policy is

$$\mathcal{L}_{\text{IL}}(\mathcal{D}, \theta, \phi) = \sum_{i=0}^{N} \sum_{t=0}^{H^i} -\log(\pi_\theta(\mathbf{a}_t^i | \mathbf{o}_t^i, g_\phi(\mathbf{o}^{\text{ref}i}_0, \mathbf{o}^{\text{ref}i}_T) - g_\phi(\mathbf{o}_0^i, \mathbf{o}_t^i))).$$

We also introduce a regularization loss function to improve compositionality by enforcing that the sum of the embeddings of two parts of a trajectory is close to the embedding of the full trajectory. We denote this a homomorphism loss $\mathcal{L}_{\text{Hom}}$ because it constrains the embedding function $g$ to preserve a mapping between concatenation of trajectories and addition of real-valued vectors. We implement the loss using the triplet margin loss from [34] with a margin equal to 1:

$$l_{\text{tri}}(a, p, n) = \max\{||a - p||_2 - ||a - n||_2 + 1.0, 0\}$$

$$\mathcal{L}_{\text{Hom}}(\mathcal{D}, \phi) \sum_{i=0}^{N} \sum_{t=0}^{H^i} l_{\text{tri}}(g_\phi(\mathbf{o}_0^i, \mathbf{o}_t^i) + g_\phi(\mathbf{o}_t^i, \mathbf{o}_T^i), g_\phi(\mathbf{o}_0^i, \mathbf{o}_T^i), g_\phi(\mathbf{o}_0^j, \mathbf{o}_T^j))$$

Finally, we follow James et al. in regularizing embeddings of paired trajectories to be close in embedding space, which has been shown to improve performance on new examples [19]. This "pair" loss $\mathcal{L}_{\text{Pair}}$ pushes the embedding of a demonstration to be similar to the embedding of its reference trajectory and different from other embeddings, which enforces that embeddings are a function of the behavior within a trajectory rather than the appearance of a state.

$$\mathcal{L}_{\text{Pair}}(\mathcal{D}, \phi) \sum_{i=0}^{N} \sum_{t=0}^{H^i} l_{\text{tri}}(g_\phi(\mathbf{o}_0^i, \mathbf{o}_T^i), g_\phi(\mathbf{o}^{\text{ref}i}_0, \mathbf{o}^{\text{ref}i}_T), g_\phi(\mathbf{o}^{\text{ref}j}_0, \mathbf{o}^{\text{ref}j}_T)$$

for any $j \neq i$. We empirically evaluate whether how these losses affect the composability of learned embeddings. While $\mathcal{L}_{\text{Pair}}$ leverages the supervision from the reference trajectories, $\mathcal{L}_{\text{Hom}}$ is entirely self-supervised.

**Measuring compositionality.** To evaluate whether the representation learned by $g$ is compositional, we condition the policy on the sum of plan vectors from multiple tasks and measure the policy's success rate. Given two reference trajectories $\mathbf{O}^{\text{ref}i}_{0:T^i}$ and $\mathbf{O}^{\text{ref}j}_{0:T^j}$, we condition the policy on $g_\phi(\mathbf{o}^{\text{ref}i}_0, \mathbf{o}^{\text{ref}i}_{T^i}) + g_\phi(\mathbf{o}^{\text{ref}j}_0, \mathbf{o}^{\text{ref}j}_{T^j})$. The policy is success if it accomplishes both tasks. We also evaluate whether the representation generalizes to more complex tasks.

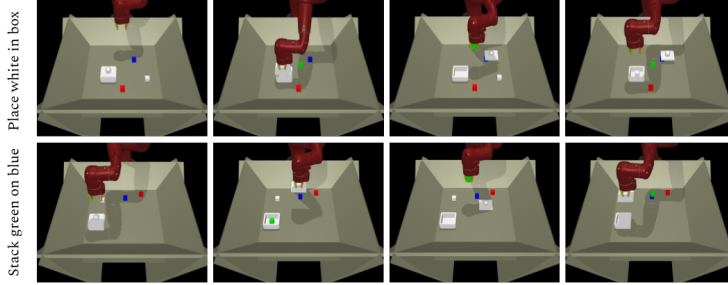

Figure 4: Two example skills from the pick and place environment. Time evolves from left to right. If the relevant objects are in the box, the agent must first remove the lid to interact with the object and also return the lid to the box in order to complete a task.

# 4 Sequential Multitask Environments

We introduce two new learning environments, shown in Figures 3 and 4, that test an agent's ability to perform tasks that require different sequences and different numbers of sub-skills. We designed these environments such that the actions change the environment and make new sub-goals possible: in the 3D environment, opening a box and removing its contents makes it possible to put something else into the box. In the environment, chopping down a tree makes it is possible to build a house. Along with the environments, we will release code to generate demonstrations of the compositional tasks.

## 4.1 Crafting Environment

The first evaluation domain is a discrete-action world where objects can be picked up and modified using tools. The environment contains 7 types of objects: tree, rock, logs, wheat, bread, hammer, axe. Logs, hammers, and axes can be picked up, and trees and rocks block the agent. The environment allows for 6 actions: up, down, left, right, pickup, and drop. The transitions are deterministic, and only one object can be held at a time. Pickup has no effect unless the agent is at the same position as a pickupable object. Drop has no effect unless an object is currently held. When an object is held, it moves with the agent. Unlike the Malmo environment which runs a full game engine [20], this environment can be easily modified to add new object types and interaction rules. We define 5 skills within the environment, *ChopTree*, *BreakRock*, *BuildHouse*, *MakeBread*, and *EatBread*, as illustrated in Figure 3. A task is defined by a list of skills. For example, a task with 3 skills could be [*ChopTree*, *ChopTree*, *MakeBread*]. Thus, considering tasks that use between 1 and 4 skills with replacement, there are 125 distinct tasks and about 780 total orderings. Unlike in Andreas et al. [1], Oh et al. [31], skill list labels are only used for data generation and evaluation; they are not used for training and are not provided to the model. The quantities and positions of each object are randomly selected at each reset.The observation space is a top-down image view of the environment, as shown in Figure 6a.

## 4.2 3D Pick and Place Environment

The second domain is a 3D simulated environment where a robot arm can pick up and drop objects. Four cubes of different colors, as well as a box with a lid, are randomly placed within the workspace. The robot's action space is a continuous 4-dimensional vector: an $(x, y)$ position at which to close the gripper and an $(x, y)$ position at which to open the gripper. The $z$ coordinate of the grasp is chosen automatically. The observation space is a concatenation of the $(x, y, z)$ positions of each of the 4 objects, the box, and the box lid. We define 3 families of skills within the environment: *PlaceInCorner*, *Stack*, and *PlaceInBox*, each of which can be applied on different objects or pairs of objects. Considering tasks that use 1 to 2 skills, there are 420 different tasks. An example of each skill is shown in Figure 4.

# 5 Experiments

Our experiments aim to understand how well CPVs can learn tasks of varying complexity, how well they can generalize to tasks that are more complex than those seen during training (thus

demonstrating compositionality), and how well they can handle additive composition of tasks, where the policy is expected to perform both of the tasks in sequence. We hypothesize that, by conditioning a policy on the subtraction of the current progress from the goal task embedding, we will learn a task representation that encodes tasks as the sum of their component subtasks. We additionally evaluate how regularizing objectives improve generalization and compositionality.

**Implementation.** We implement $g_\phi$ and $\pi_\theta$ as neural networks. For the crafting environment, where the observations are RGB images, we use the convolutional architecture in Figure 5. The encoder $g$ outputs a 512 dimensional CPV. The policy, shaded in red, takes the subtraction of CPVs concatenated with features from the current observation and outputs a discrete classification over actions.

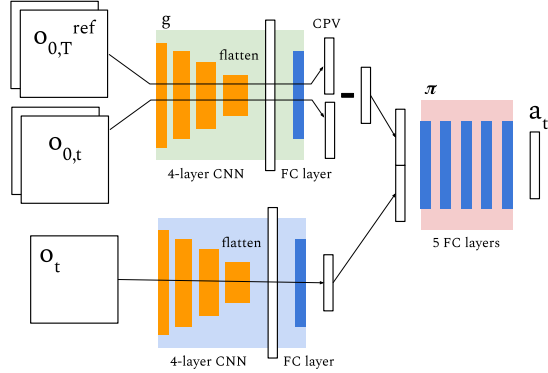

Figure 5: The network architecture used for the crafting environment. Orange denotes convolutional layers and dark blue denotes fully connected layers. The trajectories $(\mathbf{o}_0^{\text{ref}}, \mathbf{o}_T^{\text{ref}})$ and $(\mathbf{o}_0, \mathbf{o}_t)$ are each passed through $g$ (the pale green box) independently, but with shared weights. The current observation $\mathbf{o}_t$ is processed through a separate convolutional network before being concatenated with the vector $g(\mathbf{o}_0^{\text{ref}}, \mathbf{o}_T^{\text{ref}}) - g(\mathbf{o}_0, \mathbf{o}_t)$.

For the 3D environment, the observation is a state vector containing the positions of each object in the scene, including the box and box lid. The function $g$ again concatenates the inputs, but here the network is fully connected, and the current observation is directly concatenated to the subtraction of the CPVs. To improve the performance of all models and comparisons, we use an object-centric policy inspired by Devin et al. [10], where the policy outputs a softmaxed weighting over the objects in the state. The position of the most attended object is output as the first coordinates of the action (where to grasp). The object attention as well as the features from the observation and CPVs are passed to another fully connected layer to output the position for placing.

**Data generation.** For the crafting environment, we train all models on a dataset containing 40k pairs of demonstrations, each pair performs the same task. The demonstrations pairs are not labeled with what task they are performing. The tasks are randomly generated by sampling 2-4 skills with replacement from the five skills listed previously. A planning algorithm is used to generate demonstration trajectories. For the 3D environment, we collect 180k trajectories of tasks with 1 and 2 skills. All models are trained on this dataset to predict actions from the environment observations shown in Figure 6a. For both environments, we added $10\%$ of noise to the planner's actions but discarded any trajectories that were unsuccessful. The data is divided into training and validation sets 90/10. To evaluate the models, reference trajectories were either regenerated or pulled from the validation set. Compositions of trajectories were never used in training or validation.

**Comparisons.** stead consistently use the same name as you do in the environment description We compare our CPV model to several one-shot imitation learning models. All models are based on Equation 2, where the policy is function of four images: $\mathbf{o}_0, \mathbf{o}_t, \mathbf{o}_0^{\text{ref}}, \mathbf{o}_T^{\text{ref}}$. The *naïve* baseline simply concatenates the four inputs as input to a neural network policy. The TECNets baseline is an implementation of task embedding control networks from [19], where the embeddings are normalized to a unit ball and a margin loss is applied over the cosine distance to push together embeddings of the same task. The policy in TECNets is conditioned on the static reference embedding rather than the subtraction of two embeddings. For both TECNets and our model, $g$ is applied to the concatenation of the two input observations.

We perform several ablations of our model, which includes the CPV architecture (including the embedding subtraction as input the policy), the homomorphism regularization, and the pair regularization. We compare the plain version of our model, where the objective is purely imitation learning, to versions that use the regularizations. *CPV-Plain* uses no regularization, *CPV-Pair* uses only $\mathcal{L}_{\text{Pair}}$, *CPV-Hom* uses only $\mathcal{L}_{\text{Hom}}$, and *CPV-Full* uses both. To ablate the effect of the architecture vs the regularizations, we run the same set of comparisons for a model denoted TE (task embeddings) which has the same architecture as TECNets without normalizing embeddings. These experiments find

Table 1: **Evaluation of generalization and compositionality in the craftin environment.** Policies were trained on tasks using between 1 and 4 skills. We evaluate the policies conditioned on reference trajectories that use 4, 8, and 16 skills. We also evaluate the policies on the composition of skills: "2, 2" means that the embeddings of two demonstrations that each use 2 skills were added together, and the policy was conditioned on this sum. For the naïve model, we instead average the observations of the references, which performed somewhat better. All models are variants on the architecture in Figure 5. The max horizon is three times the average number of steps used by the expert for that length of task: 160, 280, and 550, respectively. Numbers are all success percentages.

| MODEL | 4 SKILLS | 8 SKILLS | 16 SKILLS | 1+1 | 2,2 | 4,4 |
|---|---|---|---|---|---|---|
| NAIVE | $29 \pm 2$ | $9 \pm 2$ | $7 \pm 2$ | $29 \pm 10$ | $24 \pm 5$ | $5 \pm 2$ |
| TECNET | $49 \pm 11$ | $17 \pm 7$ | $7 \pm 6$ | $59 \pm 11$ | $43 \pm 11$ | $\mathbf{29} \pm 23$ |
| TE | $53 \pm 4$ | $28 \pm 2$ | $\mathbf{25} \pm 20$ | $32 \pm 1$ | $44 \pm 25$ | $18 \pm 12$ |
| TE-PAIR | $64 \pm 1$ | $31 \pm 1$ | $18 \pm 2$ | $55 \pm 3$ | $53 \pm 8$ | $21 \pm 2$ |
| TE-HOM | $50 \pm 4$ | $27 \pm 2$ | $21 \pm 1$ | $51 \pm 1$ | $52 \pm 1$ | $20 \pm 1$ |
| TE-FULL | $61 \pm 8$ | $28 \pm 8$ | $13 \pm 2$ | $60 \pm 1$ | $47 \pm 7$ | $23 \pm 7$ |
| CPV-NAIVE | $51 \pm 8$ | $19 \pm 5$ | $9 \pm 2$ | $31 \pm 16$ | $30 \pm 15$ | $5 \pm 2$ |
| CPV-PAIR | $\mathbf{68} \pm 11$ | $\mathbf{44} \pm 14$ | $\mathbf{31} \pm 13$ | $2 \pm 3$ | $1 \pm 2$ | $0 \pm 0$ |
| CPV-HOM | $63 \pm 3$ | $35 \pm 5$ | $27 \pm 8$ | $71 \pm 8$ | $60 \pm 11$ | $\mathbf{26} \pm 14$ |
| CPV-FULL | $\mathbf{73} \pm 2$ | $\mathbf{40} \pm 3$ | $\mathbf{28} \pm 6$ | $\mathbf{76} \pm 3$ | $\mathbf{64} \pm 6$ | $\mathbf{30} \pm 10$ |

whether the regularization losses produce compositionality on their own, or whether they work in conjunction with the CPV architecture.

Table 2: **3D Pick and Place Results.** Each model was trained on tasks with 1 to 2 skills. We evaluate the models on tasks with 1 and 2 skills, as well as the compositions of two 1 skill tasks. For each model we list the success rate of the best epoch of training. All numbers are averaged over 100 tasks. All models are variants of the object-centric architecture, shown in the supplement. We find that the CPV architecture plus regularizations enable composing two reference trajectories better than other methods.

| MODEL | 1 SKILL | 2 SKILLS | 1,1 |
|---|---|---|---|
| NAIVE | $65 \pm 7$ | $34 \pm 8$ | $6 \pm 2$ |
| TECNET | $82 \pm 6$ | $50 \pm 2$ | $33 \pm 4$ |
| TE-PLAIN | $\mathbf{91} \pm 2$ | $55 \pm 5$ | $22 \pm 2$ |
| TE-PAIR | $81 \pm 11$ | $51 \pm 8$ | $15 \pm 3$ |
| TE-HOM | $\mathbf{92} \pm 1$ | $\mathbf{59} \pm 1$ | $24 \pm 12$ |
| TE-FULL | $88 \pm 2$ | $55 \pm 8$ | $9 \pm 6$ |
| CPV-PLAIN | $87 \pm 2$ | $55 \pm 2$ | $52 \pm 2$ |
| CPV-PAIR | $82 \pm 4$ | $42 \pm 3$ | $7 \pm 1$ |
| CPV-HOM | $88 \pm 1$ | $54 \pm 5$ | $\mathbf{55} \pm 4$ |
| CPV-FULL | $87 \pm 4$ | $54 \pm 4$ | $\mathbf{56} \pm 6$ |

**Results.** We evaluate the methods on both domains. To be considered successful in the crafting environment, the agent must perform the same sub-skills with the same types of objects as those seen in the reference trajectory. The results on the crafting environment are shown in Table 1, where we report the mean and standard deviation across 3 independent training seeds. We see that both the naïve model and the TECNet model struggle to represent these complex tasks, even the 4 skill tasks that are in the training distribution. We also find that both the CPV architecture and the regularization losses are necessary for both generalizing the longer tasks and composing multiple tasks. The pair loss seems to help mostly with generalization, while the homomorphism losses helps more with compositionality. CPVs are able to generalize to 8 and 16 skills, despite being trained on only 4 skill combinations, and achieve 76% success at composing two tasks just by adding their embedding vectors. Recall that CPVs are not explicitly *trained* to compose multiple reference trajectories in this way – the compositionality is an extrapolation from the training. The TE ablation, which does not use the subtraction of embeddings as input to the policy, shows worse compositionality than our method even with the homomorphism loss. This supports our hypothesis that structural constraints over the embedding representation contribute significantly to the learning.

These trends continue in the pick and place environment in Table 2, were we report the mean and standard deviation across 3 independent training seeds. In this environment, a trajectory is successful if the objects that were moved in the reference trajectory are in the correct positions: placed in each corner, placed inside the box, or stacked on top of a specific cube. As expected, TECNet performs well on 1 skill tasks which only require moving a single object. TECNet and the naïve model fail to compose tasks, but the CPV model performs as well at composing two 1-skill tasks as it does when imitating 2-skill tasks directly. As before, the TE ablation fails to compose as well as CPV, indicating that that the architecture and losses together are needed to learned composable embeddings.

## 6   Discussion

Many tasks can be understood as a composition of multiple subtasks. To take advantage of this latent structure without subtask labels, we introduce the compositional plan vector (CPV) architecture along with a homomorphism-preserving loss function, and show that this learns a compositional representation of tasks. Our method learns a task representation and multi-task policy jointly. Our main idea is to condition the policy on the arithmetic difference between the embedding of the goal task and the embedding of the trajectory seen so far. This constraint ensures that the representation space is structured such that subtracting the embedding of a partial trajectory from the embedding of the full trajectory encodes the portion of the task that remains to be completed. Put another way, CPVs encode tasks as a set of subtasks that the agent has left to perform to complete the full task. CPVs enable policies to generalize to tasks twice as long as those seen during training, and two plan vectors can be added together to form a new plan for performing both tasks.

We evaluated CPVs in a one-shot imitation learning setting. Extending our approach to a reinforcement learning setting is a natural next step, as well as further improvements to the architecture to improve efficiency. A particularly promising future direction would be to enable CPVs to learn from unstructured, self-supervised data, reducing the dependence on hand-specified objectives and reward functions.

## 7   Acknowledgements

We thank Kate Rakelly for insightful discussions and Hexiang Hu for writing the initial version of the 3D simulated environment. This material is based upon work supported by the National Science Foundation Graduate Research Fellowship Program under Grant No. DGE 1752814.

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

# A Network Architectures

## A.1 Crafting environment

The observation is an RGB image of 33x30 pixels. The architecture for $g$ concatenates the first and last image of the reference trajectory along the channel dimension, to obtain an input size of 33x30x6. This is followed by 4 convolutions with 16, 32,64, and 64 channels, respectively, with ReLU activations. The 3x3x64 output is flattened and a fully connected layer reduces this to the desired embedding dimension. The same architecture is used for the TECNet encoder. For the policy, the observation is passed through a convolutional network with the same architecture as above and the output is concatenated with the subtraction of embeddings as defined in the paper's method. This concatenation is passed through a 4 layer fully connected network with 64 hidden units per layer and ReLU activations. The output is softmaxed to produce a distribution over the 6 actions. The TECNet uses the same architecture, but the reference trajectory embeddings are normalized there is no subtraction; instead, the initial image of the current trajectory is concatenated with the observation. The naive model uses the same architecture but all four input images are concatenated for the initial convolutional network and there is no concatenation at the embedding level.

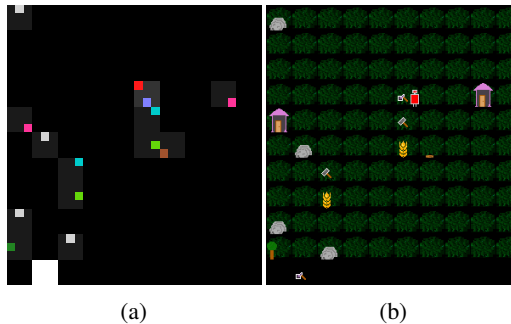

(a)           (b)

Figure 6: The crafting environment. (a) Shows a state observation as rendered for the agent. The white square in the bottom left indicates that an object is held by the agent. (b) Shows the same state, but rendered in a human-readable format. The axe shown in the last row indicates that the agent is currently holding an axe.

## A.2 3D environment

The environment has 6 objects: 4 cubes (red, blue, green, white), a box body and a box lid. The state space is the concatenation of the $(x, y, z)$ positions of these objects, resulting in an 18-dimensional state. As the object positions are known, we use an attention over the objects as part of the action, as shown in Figure 7. The actions are 2 positions: the $(x_0, y_0)$ position at which to grasp and the the $(x_1, y_1)$ position at which to place. When training the policy using the object centric model, $(x_0, y_0)$ is a weighted sum of the object positions, with the $z$ coordinate being ignored. Weights over the 6 object are output by a neural network given the difference of CPVs and the current observation. At evaluation time, $(x_0, y_0)$ is the $\arg\max$ object position. This means that all policies will always grasp at an object position. For $(x_1, y_1)$, we do not have the same constraint. Instead, the softmaxed weights over the objects are concatenated with the previous layer's activations, and another fully connected layer maps this directly to continuous valued $(x_1, y_1)$. This means that the policy can place at any position in the workspace. The naïve model, TECNet model, and CPV models all use this object-centric policy, then only differ in how the input to the policy.

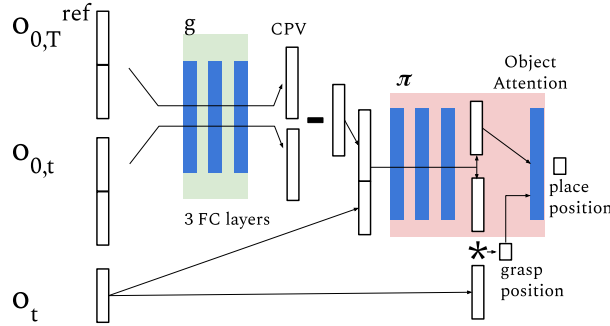

Figure 7: The object-centric network architecture we use for the 3D grasping environment. Because the observations include the concatenated positions of the objects in the scene, the policy chooses a grasp position by predicting a discrete classification over the objects grasping at the weighted sum of the object positions. The classification logits are passed back to the network to output the position at which to place the object.

## B   Hyperparameters

We compared all models across embedding dimension sizes of [64,128,256, and 512]. In the crafting environment, the 512 size was best for all methods. In the grasping environment, the 64 size was best for all methods. For TECNets, we tested $\lambda_{\text{ctr}} = 1$ and 0.1, and found that 0.1 was best. All models are trained on either k-80 GPUs or Titan X GPUs.

## C   Additional Experiments

We ran a pared down experiment on a ViZDoom environment to show the method working from first person images, as shown in C. In the experiment, the skills are reaching 4 different waypoints in the environment. The actions are "turn left," "turn right," and "go forward." The observation space consists of a first person image observation as well as the $(x, y)$ locations of the waypoints. We evaluate on trajectories that must visit 1 or 2 waypoints (skills), and also evaluate on the compositions of these trajectories. The policies were only trained on trajectories that visit up to 3 waypoints. These evaluations are shown in 3.

Figure 8: First person view in VizDoom env.

Table 3: **ViZDoom Navigation Results.** All numbers are success rates of arriving within 1 meter of each waypoint.

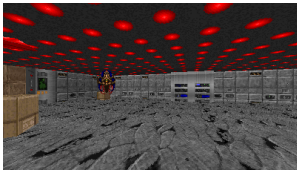

| MODEL | 1 SKILL | 2 SKILLS | 1+1 | 2+2 |
|-------|---------|----------|------|-----|
| NAIVE | 97 | 94 | 36.7 | 2 |
| TECNET | 96 | 95.3 | 48.3 | 0 |
| CPV | 93 | 90.7 | **91** | **64** |

