[Reviews · NeurIPS 2019]

Reviewer 1



It would be interesting to increase the complexity of the tasks. Also, I'd love to see a comparison to other related methods and an explanation of why you're not using language to describe the tasks, e.g. https://arxiv.org/abs/1712.07294

Reviewer 2



The authors propose a new method called compositional plan vectors to handle complex tasks which involve compositions of skills. The method is based on a simple but effective idea of using the addition of task representation vectors to compose tasks. In my opinion, learning complex tasks involving many subskills is an important and challenging problem and the proposed method provides a novel and interesting way to approach this problem. The paper is written well and easy to follow. The authors position their method well with respect to prior work. The experiments test the proposed model in 2 diverse simulation environments. The proposed model is shown to perform better than a competitive baseline in both environments. CPV also outperform the baselines in generalizing to unseen longer tasks. One of my concerns about the submissions is lack of motivation behind introducing new environments. Using existing environments make it easier to compare with prior methods and help in reproducibility. The authors should justify the need for new environments.

Reviewer 3



Summary The paper proposes a new method for better and more efficient generalization to more complex tasks at test time in the setting of one-shot imitation learning. The main idea is to condition the policy on the difference between the embedding of some reference trajectory and the a partial trajectory of the agent (for the same task, but starting from a potentially different state of the environment). Main Comments I found the experimental section to be slightly thin and I would like to see how this method performs on at least another more complex task. It would also be good to include a discussion on the types of environments where we can expect this to perform best and where we can expect it to fail or perform worse than other relevant algorithms. I also think more comparisons with other approaches for one-shot imitation learning (such as Duan et al. 2017) are needed for strengthening the paper. How does CPV compare to other imitation learning algorithms such as Behavioral Cloning, Dagger, or GAIL? I believe that the paper’s related work section could be significantly improved. It seems to me that it is lacking some important references on learning using task embeddings / goal representations, as well as works on multi-task learning, transfer learning and task compositionality. Some examples of relevant references are: C. Devin, A. Gupta, T. Darrell, P. Abbeel, and S. Levine. Learning modular neural network policies for multi-task and multi-robot transfer. In ICRA, 2017. B. C. da Silva, G. Konidaris, and A. G. Barto. Learning parameterized skills. In ICML, 2012. T. Schaul, D. Horgan, K. Gregor, and D. Silver. Universal value function approximators. In ICML, 2015. S. P. Singh. The efficient learning of multiple task sequences. In NIPS, 1991 Haarnoja et al 2018, Composable Deep Reinforcement Learning for Robotic Manipulation Hausman et al. 2018, Learning an embedding space for Transferable Robot Skills Minor Comments: You mention that the TECNet normalizes the embedding. Is the embedding normalized in the same way in your model? It would be good to add some experiments to tease apart the effects of this normalization. How much of the difference in the performance of CPV and that of the TECNet is due to this normalization? Some details regarding the state representation, the model architecture and training parameters are missing It would be interesting to see if this method can be applied in RL contexts as well and how well it generalizes to new tasks. Clarity The paper is clearly written overall. Significance The problem of generalizing to unseen tasks is a very important and unsolved one in the field. The paper seems relevant for the community although I believe it could be applied outside of imitation learning as well. While the paper doesn’t introduce a completely novel concept or approach, the way it combines the different building blocks seems promising for obtaining a better performing method for one-shot imitation learning and generalization to unseen tasks. I believe people in the community would learn something new from reading the paper and could build on its findings. Originality The paper doesn't introduce a fundamentally novel idea, but the way it combines previous ideas is not entirely trivial and the authors demonstrate its benefits compared to relevant ablations. Quality The paper seems technically correct to me. ---------- UPDATE: I have read the rebuttal and the other reviews. The authors addressed most of my concerns, added comparisons with a strong baseline and evaluated their approach on another environment. The results show that their method is able do well on certain compositional tasks on which the baseline fails to gain any reward. For all these reasons, I have increased my score by 1 point (total of 7).

[Author Response · NeurIPS 2019]

We thank all the reviewers for their helpful feedback and positive view of our work. To address the reviewers concerns (**R3,R4,R5**), we have added a comparison to Duan et al.'s One Shot Imitation learning in Tables 3 and 2, a comparison to a non-normalized TECNet ablation, as well as an evaluation on a VizDoom navigation task in Table 1. We believe that these additions address all of the main reviewer concerns.

**R3,R5** *Lacking some important references on learning using task embeddings / goal representations, etc* Thank you for the pointers to more related work; missing these was an oversight. Hausman et al. is discussed at line 95 of the paper, and we will add discussion of the additional papers, as well as the language HRL paper from R3, in the related work section. Combining CPVs with natural language task descriptions is an interesting avenue for future work.

**R5** *TECNet normalizes the embedding. Is the embedding normalized in the same way in your model?* We do not normalize the CPV embeddings. To tease apart the effects of normalization, we have added a comparison to non-normalized versions of TECNet, which are labeled "TE" (task embedding) in Table 3 and 2.

**R4,R5** *More Comparisons.* We have added a comparison to Duan et al.'s One shot imitation method. As the authors have not released an implementation of the method or environment, we implemented the key details of the method: the reference demonstration is encoded with a residual 1D convolution, and the LSTM policy attends over the reference trajectory. This and the TE comparison will be run on the VizDoom env for the camera ready version.

**R4, R5** *Lack of motivation behind introducing new environments.* We agree that benchmark environments are ideal for the integrity of the field. To address this, we have added a navigation task from VizDoom. Unfortunately, most currently available environments are too simple to benefit from a compositional representation of tasks. The environment in Duan et al. was not made public. The environment from Sohn et al. was only released this summer (after the deadline). We are releasing our environments publicly with documentation, training code, and demonstration data.

**R5** *How does CPV compare to other imitation learning algorithms such as Behavioral Cloning, Dagger, or GAIL?* CPV can be used in conjunction with any imitation learning algorithm. In our results we use behavioral cloning, and we plan to try IRL methods such as GAIL in future work.

Table 1: **ViZDoom Navigation Results.** We evaluate our method in ViZDoom where the goal is to visit waypoints in a predetermined order. The actions are "turn left," "turn right," and "go forward." The observation space consists of a first person image observation as well as the locations of the waypoints. We evaluate on trajectories that must visit 1 or 2 waypoints (skills), and also evaluate on the compositions of these trajectories. The policies were only trained on trajectories that visit up to 3 waypoints. All numbers are success rates of arriving within 1 meter of each waypoint.

| MODEL | 1 SKILL | 2 SKILLS | 1+1 | 2+2 |
|---|---|---|---|---|
| NAIVE | 97 | 94 | 36.7 | 2 |
| TECNET | 96 | 95.3 | 48.3 | 0 |
| CPV | 93 | 90.7 | **91** | **64** |

Table 2: **3D Pick and Place Results.** We added the TE and TE-Pair baselines, which use a task embedding like TECNet but without normalizing to the unit ball. TE-Pair has a triplet margin loss so that embeddings of the same task should be close in feature space, which is like the TECNet margin loss. The plain TE only uses the imitation learning loss. While TE performs well at the training tasks in this environment, it does not succeed at compositions of tasks. The Duan et al. architecture fails in this environment.

| MODEL | 1 SKILL | 2 SKILLS | 1,1 |
|---|---|---|---|
| TECNET | $82 \pm 6$ | $50 \pm 2$ | $33 \pm 4$ |
| CPV | $87 \pm 2$ | $\mathbf{55} \pm 2$ | $\mathbf{52} \pm 2$ |
| TE | $\mathbf{91} \pm 2$ | $\mathbf{55} \pm 5$ | $22 \pm 2$ |
| TE-PAIR | $81 \pm 11$ | $51 \pm 8$ | $15 \pm 3$ |
| DUAN ET AL. | $6 \pm 1$ | $0 \pm 0$ | $0 \pm 0$ |

Figure 1: First person view in VizDoom env. The agent must navigate through multiple waypoints.

Table 3: **2D crafting results.** The TE ablation, which is like TECNet but with un-normalized embeddings performs worse than TECNet. The Duan et al. architecture performs well in this crafting environment.

| Model | 4 Skills | 8 Skills | 16 Skills | 2,2 | 2,2,2,2 | 4,4 |
|---|---|---|---|---|---|---|
| TECNet | $50 \pm 14$ | $39 \pm 5$ | $8 \pm 11$ | $52 \pm 15$ | $17 \pm 5$ | $34 \pm 22$ |
| CPV | $65 \pm 10$ | $\mathbf{80} \pm 3$ | $44 \pm 9$ | $55 \pm 5$ | $29 \pm 9$ | $\mathbf{58} \pm 8$ |
| CPV-Hom. | $\mathbf{84} \pm 12$ | $82 \pm 15$ | $54 \pm 8$ | $\mathbf{71} \pm 1$ | $29 \pm 10$ | $48 \pm 14$ |
| TE | $29 \pm 4$ | $21 \pm 29$ | $3 \pm 2$ | $35 \pm 10$ | $20 \pm 12$ | $15 \pm 7$ |
| TE-Pair | $25 \pm 6$ | $12 \pm 2$ | $0 \pm 0$ | $36 \pm 9$ | $10 \pm 3$ | $12 \pm 4$ |
| Duan et al. | $75$ | $67$ | $\mathbf{80}$ | $59$ | $\mathbf{62}$ | $\mathbf{66}$ |

[Meta-Review · NeurIPS 2019]

The one reviewer on the fence was convinced by the rebuttal. All reviewers consider the method interesting and the experiments sufficient for demonstrating the value of the method.